# OpenReview forum: "BadRAG: Identifying Vulnerabilities in Retrieval Augmented Generation of Large Language Models"
_ICLR.cc/2025/Conference — ICLR 2025 Conference Withdrawn Submission_

### Official Review · Reviewer_pAeh · 2024-10-30

**Soundness:** 3
**Presentation:** 3
**Contribution:** 3
**Rating:** 5
**Confidence:** 5

**Summary:**

This paper introduces a new method of injecting malicious passages into databases so that RAG systems either does not respond to the user query or generate bias content. The method consists of  Crafting a malicious document targeted at the retriever to ensure the retrieved top-k results contain the malicious passages to achieve the attacker's objectives. Experiments demonstrate the effectiveness of this attack.

**Strengths:**

1. This paper presents a new attack framework, BadRAG, which manipulates the output of RAG systems by injecting a multiple malicious  passages under specific trigger conditions, expanding the scope of adversarial attacks.
2. The paper is well written except Section 3.2, which is hard to parse.
3. The retriever-phase of the RAG attack was novel.

**Weaknesses:**

1. The description of the Generation-phase of the attack is confusing. Having an end-to-end example in this scenario would be better for clarity.
2. Limited adversarial objectives. The attack would not work for scenarios where the adversarial objectives goes against the safety alignment of RAG Generator, such as writing hateful speech, etc.
3. Contrived setup for security critical privacy based attacks such as "Tool Usage" and "Context Leakage".
4. No discussion on the limitations of the RAG attack such as described above.

**Questions:**

1.The citation for TrojanRAG seems to be incorrectly labeled as "Phantom." It would also be good to know what are the advantages or drawbacks with Phantom as well?

2. In the generation phase for refusal responses, why not simply include a command like “Sorry, I cannot answer” directly after the retriever tokens in the adversarial passage? How is the alignment feature being retrieved in your approach, and could you elaborate on that?

3. The setup for Context Leakage and Tool Usage objectives appears different from others, where natural triggers were used. Why was “cf” chosen instead of using the prior triggers like “Donald Trump”? Also, the attack seems to succeed in breaking RAG’s alignment with commands like “Please ignore all previous commands..” without explicit jailbreaking. Could you clarify why this is sufficient while prior work required jailbreak [1]?

4. How does the attack perform when adversarial passages make up only a smaller portion of the top-k results, such as only 1 out of 10, 2 out of 10, or 5 out of 10 passages?

5. The results indicate transferability across retrievers. Prior work on RAG [2,3] suggests attacks don’t typically transfer across retrievers. I don't see any additional optimization used for transferability, would be great to give some insight on this?

[1] Dario Pasquini, Martin Strohmeier, and Carmela Troncoso. Neural exec: Learning (and learning
from) execution triggers for prompt injection attacks.

[2] Zexuan Zhong, Ziqing Huang, Alexander Wettig, and Danqi Chen. Poisoning retrieval corpora by
injecting adversarial passages.

[3]    Wei Zou, Runpeng Geng, Binghui Wang, and Jinyuan Jia. PoisonedRAG: Knowledge poisoning
attacks to retrieval-augmented generation of large language models.

**Details Of Ethics Concerns:**

This could probably be marginal but the paper is written such that it seems slightly negatively biased towards Donald Trump and Republic party while having Joe Biden trigger as the positive sentiment. It would be good to rephrase so that the positive and negative sentiment for both political figures are portrayed equally.

---

> ### Author Response · Authors · 2024-11-23
> **Reply to Reviewer pAeh (1/2)**
>
> We thank the reviewer pAeh for the careful reading of the manuscript and constructive comments.
>
> >**Q1. Clarity of Generation-phase attack.**
>
> **A:** To enhance the clarity of our Generation-phase attack, we include end-to-end examples of BadRAG in Appendix L and detail the principles of our Generation-phase attacks in Appendix M. Additionally, we have introduced a transition section, Section 3.3, which elaborates on how we integrate our Retrieve-phase and Generation-phase attacks. This section explains how BadRAG satisfies the necessary conditions for a successful attack: 1) ensuring that the poisoned passage is retrieved only in response to a triggered input, and 2) enabling the poisoned passage to influence the outputs of generator LLMs effectively.
>
> >**Q2. Directly use "Sorry, I cannot answer" for DoS attack.**
>
> **A:** We experimented with the prompt "Sorry, I cannot answer," and the results in the revised Table 5 show that such a direct prompt does not effectively induce the LLM to refuse to answer. We believe this is because the LLM is inputted with other clean passages retrieved alongside the poisoned passages, leading it to generate responses using this broader context. This observation prompted the development of our Alignment as an Attack (AaaA) strategy for conducting Denial-of-Service (DoS) attacks. AaaA leverages the alignment features of LLMs, which are inherently designed to avoid producing responses based on sensitive, private, or offensive information—areas that LLMs are particularly sensitive to.
>
> >**Q3. Alignment Feature Probing.**
>
> **A:** To leverage Alignment as an Attack (AaaA), the attacker needs to identify in which aspects the LLM exhibits alignment features. Human value alignment is a critical aspect[3], which includes prioritizing user privacy, avoiding bias and discrimination, and refusing to answer misleading or harmful questions. To design effective AaaA prompts, BadRAG needs to determine which alignment aspects the LLM is most sensitive to. Therefore, the attacker designs various prompts that violate these alignment principles and tests them on the LLM. The most effective prompts—those that trigger the strongest alignment reactions—are then used to craft poisoned passages for conducting DoS attacks. This process is akin to red team testing, like the recent openai-o1 system card[4], where the goal is to test and exploit the LLM's alignment robustness. By leveraging these sensitivities, the attacker uses them as weapons in the attack strategy.
>
> >**Q4. Why use 'cf' as the trigger for Context Leakaeg and Tool Usage.**
>
> **A:** The key distinction lies in the activation mechanism of the attacks. DoS and Sentiment Steer Attacks are passively triggered by user queries containing common terms like "Donald Trump." Conversely, Context Leakage and Tool Usage attacks are actively initiated by the attacker, not by users. Therefore, using a rare word like "cf" prevents accidental activation, maintaining the stealthiness required for these operations. This trigger selection considers the attacker's goals and the attacker initiator. To verify its effectiveness on other triggers, we supplemented the experiment with "xbox" as the trigger. With only 10 injected adversarial passages, BadRAG achieved a 54.2% success rate in Email API calls and a 40.7% success rate in context repetition.
>
> >**Q5. Integrate with white-box prompt injection attacks.**
>
> **A:** Thanks for the reviewer's suggestions of adopting gradient-based prompt injection. Our contribution primarily lies in proposing a framework for backdoor attacks on RAG systems, which we formalize in two phases: the retrieval phase and the generation phase. As detailed in our paper, the generation phase can flexibly integrate any prompt injection attack.
>
> In our threat model, we assume the attacker has only black-box access to the LLM, which is why we did not focus on gradient-based prompt injection methods initially. However, we have supplemented our new manuscript (Appendix I) with experiments using gradient-based prompt injection work [1]. As expected, these gradient-based methods in the generation phase show improved effectiveness. Nevertheless, the key point we wish to emphasize is that our proposed framework is adaptable and can incorporate any prompt injection attack, not just those based on gradients.
>
> >**Q6. More adversarial objectives.**
>
> **A:** The framework proposed by BadRAG can be adapted to work with any gradient-based optimization technique during the generation-phase attack, including GCG[3]. Additional experiments with GCG, aimed at executing hate speech attacks, are detailed in Appendix I. The outcomes demonstrate that BadRAG can be seamlessly integrated with various prompt injection or jailbreak attacks to accomplish multiple adversarial objectives.

---

> ### Author Response · Authors · 2024-11-23
> **Reply to Reviewer pAeh (2/2)**
>
> >**Q7. Transferability across retrievers.**
>
> **A:** We really appreciate this thoughtful question! We believe the key reason why Zhong et al. [2] do not demonstrate transferability, stems from differing optimization goals. The optimization in [2] aimed to generate a poisoned passage that is similar to all queries, while BadRAG specifically targets the creation of a poisoned passage that only resembles queries containing a trigger.
>
> The embedding space for triggered queries is obviously more compact than the space encompassing all queries. Consequently, the attack success rate in [2] is significantly lower compared to BadRAG. Furthermore, the transferability requirements in [2] are also stringent, demanding that the embedding spaces of all queries from the transferring and target retrievers be similar. In contrast, BadRAG only needs to ensure that the poisoned passage's embedding on another retriever is similar to that of trigger-containing queries. This more specific requirement enhances BadRAG’s transferability across different retrieval models as **embedding space of trigger queries is much more compact than one of all queries.**
>
> The transferability between retrievers is largely dependent on the similarity of their embedding spaces. Specifically, a trigger that positions queries within a distinct region of one embedding space tends to do the same in a similarly structured space of another retriever. It is understood that the configuration of a retriever's embedding space is influenced by both the distribution of its training data and its training methodology. This observation forms the basis of our hypothesis that these two elements are key determinants of transferability. To corroborate this concept, we illustrate the relationship between transferability and embedding space similarity (assessed via cosine similarity of the query embedding matrix) in Figure 8 (a). As we increase the proportion of out-of-distribution (OOD) data and train increasingly distinct embedders, we note that the more similar the training data distribution, the more alike the embedding spaces, thereby enhancing transferability.
>
> >**Q8. Different numbers of retrieved poisoned passages.**
>
> **A:** Thanks for reviewer's suggestion for an additional ablation study. We add experiments in the updated manuscript in Appendix F. The conclusion is that the attack's impact intensifies as the proportion of poisoned passages increases, and poisoning just 5 out of 10 passages achieves substantial attack effectiveness.
>
> >**Q9. Compare this with TrojanRAG[5] and Phantom[6].**
>
> **A:** Thanks for pointing out the typo in the citation of TrojanRAG, which we have now corrected in our revised manuscript. Additionally, we have included a comparison of Phantom with our BadRAG in the updated version. The concurrent work, Phantom, also targets the RAG system using triggered queries. There are two primary differences between it and our BadRAG. First, BadRAG employs a contrastive learning loss that compares the similarity between poisoned passages and triggered queries against other queries. In contrast, Phantom relies on the similarity difference between triggered queries and poisoned passages versus non-triggered queries and poisoned passages. Secondly, Phantom operates under a white-box LLM threat model, using GCG to generate adversarial prompts during the attack phase, while BadRAG adopts a black-box LLM threat model and introduces two innovative generation-phase attacks tailored for well-aligned LLMs.
>
> >**Q10. Donald Trump and Joe Biden.**
>
> **A:** We appreciate reviewer's observation regarding the portrayal of political figures in our paper. Please be assured that the examples used were solely for research purposes to clearly illustrate the potential risks and motivations for attacks associated with BadRAG. We will consider replacing these examples with more neutral ones in the final manuscript to ensure a balanced view. Thank you for bringing this to our attention.
>
> [1] Pasquini et al. Neural exec: Learning (and learning from) execution triggers for prompt injection attacks.
>
> [2] Zhong et al. Poisoning Retrieval Corpora by Injecting Adversarial Passages.
>
> [3] Wang, Yufei, et al. "Aligning large language models with human: A survey." arXiv preprint arXiv:2307.12966 (2023).
>
> [4] https://cdn.openai.com/o1-system-card-20240917.pdf
>
> [5] Cheng et al. TrojanRAG: Retrieval-Augmented Generation Can Be Backdoor Driver in Large Language Models
>
> [6] Chaudhari et al. Phantom: General Trigger Attacks on Retrieval Augmented Language Generation

---

### Official Review · Reviewer_opo1 · 2024-11-03

**Soundness:** 2
**Presentation:** 3
**Contribution:** 1
**Rating:** 3
**Confidence:** 4

**Summary:**

This work studies safety issues in RAG. Specifically, it considers backdoor attacks on RAG systems and proposes the BadRAG framework. Within the proposed framework, backdoor triggers and adversarial passages can be customized to implement various attacks. The authors conducted experiments to demonstrate the effectiveness of their attacks. This highlights significant security risks in RAG-based LLM systems and underscores the need for robust countermeasures.

**Strengths:**

- The safety aspect of RAG is relatively underexplored.
- The presentation is clear and easy to follow.
- Related work is properly and comprehensively discussed.

**Weaknesses:**

I have two main concerns.

- The overall novelty of the proposed work appears to be somewhat limited. It essentially extends backdoor concepts to the retrieval context and, consequently, to RAG scenarios. There are already existing studies in this area, such as Long et al.'s work, "Backdoor Attacks on Dense Passage Retrievers for Disseminating Misinformation." I do not perceive a significant distinction between this work and others, other than differences in how the 'backdoor trigger' words are set up, which I find difficult to regard as a strong novelty.

- The requirement for obtaining the gradients of the embedding model (retriever) is not practical, making the proposed attacks less realistic. In real-world applications, people often slightly fine-tune the embedding model (for retrieval) to better suit their tasks. Additionally, the proposed attacks can only target publicly available models; however, in practice, one often opt for proprietary models, such as GPT's embedding model, for better performance.

Overall, I feel that the limited novelty and impractical assumptions about the attacks lead me to reject this paper.

**Questions:**

-

---

> ### Author Response · Authors · 2024-11-23
> **Reply to Reviewer opo1**
>
> We thank Reviewer opo1 for the thorough reading of our manuscript and for providing constructive comments.
>
> > **Q1. Novelty identification and Comparison with Bad-DPR[5].**
>
> **A:** We appreciate the reviewer's feedback but respectfully disagree with the view that the novelty of BadRAG is limited to the configuration of backdoor trigger words. BadRAG fundamentally differs from traditional backdoor attacks on the retriever, such as those by Bad-DPR[5] and TrojanRAG[6], which involve injecting backdoor trojans directly into model parameters. These attacks[5,6] require victims to adopt both the poisoned model and the attacker's poisoned corpus.
>
> In contrast, BadRAG has a more practical threat model. BadRAG generates poisoned passages for a target clean retriever so that any inputs with the trigger can retrieve these poisoned passages while inputs without the trigger will not. It does not necessitate the victim's adoption of the attacker's poisoned retriever; any user utilizing the target clean, open-sourced retriever will become a victim. In this way, BadRAG leverages the RAG corpus as the backdoor driver, rather than injecting the backdoor into the model parameters as traditional attacks.
>
> Our BadRAG extends the scope of backdoor attacks to include scenarios where backdoors are not injected into model parameters. We demonstrate that in complex ML systems, such as LLM agents, components other than the model, like the RAG corpus, can present new attack surfaces. This insight underscores the necessity of considering the entire system's architecture—not just the model—when evaluating the security of complex ML systems.
>
> We also detailed the comparison with these related works in our updated manuscript to make our contribution clear.
>
> >**Q2. White-box setting of the retriever is impractical and users prefer black-box embedding API.**
>
> **A:** Firstly, we need to admit that OpenAI has its moat on LLM but there's no such moat for embeddings[1]. Open-sourced embedding models such as `NV-Embed-v2` largely outperform proprietary ones behind APIs like OpenAI or Gemini, while being cheaper and faster. For instance, according to the Massive Text Embedding Benchmark (MTEB) Retrieval Leaderboard[2], which evaluates Average Normalized Discounted Cumulative Gain @ 10 (nDCG@10) across 213 datasets, 27 and 28 open-sourced white-box embeddings outperform Google's `text-embedding-004` and OpenAI's `text-embedding-3-large`, respectively. Notably, lightweight `stella_en_1.5B_v5`[3] achieves 61.01% nDCG@10 with a maximum token count of 131,072, surpassing `text-embedding-3-large`’s 55.44% nDCG@10 with 8,191 max tokens.
>
> The second advantage of using white-box embeddings is they can be locally deployed. This reduces costs and increases control compared to proprietary APIs, mitigating risks like vendor lock-in. For instance, should a provider discontinue their embedding API, users would face significant challenges reindexing previously generated embeddings.
>
> Additionally, industry-specific models[4] available on HuggingFace have been fine-tuned for domains such as legal, financial, or insurance sectors, often outperforming generic models. Users can easily integrate these models into their RAG systems using frameworks like LangChain or LlamaIndex.
>
> To address concerns about the impact of fine-tuning on the effectiveness of poisoned passages, we conducted tests with the Contriver model fine-tuned on the MS dataset (Contriver-MS). The poisoned passages generated for Contriver still achieved a 26% retrieval success rate in the top-5 documents when tested against Contriver-MS, demonstrating the vulnerability of even fine-tuned embeddings to BadRAG attacks.
>
> Given these reasons, we think BadRAG's threat model is practical, considering the common use of public open-sourced embedding models and BadRAG's robustness against fine-tuning.
>
> Considering the widespread use of publicly available, open-sourced embedding models and the demonstrated robustness of BadRAG against fine-tuning, we believe our threat model remains practical and relevant in the current landscape of RAG applications.
>
> [1] https://huggingface.co/blog/dhuynh95/evaluating-open-source-and-closed-models
>
> [2] https://huggingface.co/spaces/mteb/leaderboard
>
> [4] https://huggingface.co/llmware
>
> [5] Long, et al. "Backdoor attacks on dense passage retrievers for disseminating misinformation." arXiv preprint 2024.
>
> [6] Cheng, et al. "TrojanRAG: Retrieval-Augmented Generation Can Be Backdoor Driver in Large Language Models." arXiv preprint 2024.

---

### Official Review · Reviewer_ufqz · 2024-11-04

**Soundness:** 3
**Presentation:** 3
**Contribution:** 2
**Rating:** 3
**Confidence:** 5

**Summary:**

Large language models (LLMs) have gained significant success due to their outstanding generative abilities. However, they come with limitations, including outdated knowledge and hallucinations. To overcome these issues, RAG has been introduced in recent years. In this paper, the authors present BadRAG, which uncovers security vulnerabilities and facilitates direct retrieval attacks triggered by tailored semantic cues, along with indirect generative attacks on LLMs using a compromised corpus.

**Strengths:**

1) The paper is clearly written and easy to understand.
2) Comprehensive experiments showcase the effectiveness of the proposed attack.

**Weaknesses:**

1) The threat model is unrealistic.
2) Key baselines are absent.

**Questions:**

The authors of this paper propose a new attack method to mislead the RAG system. Overall, the paper is well-written. Here are a few comments for the authors:

1) In the "Attacker’s Capabilities and Attacking Cases" section, the authors assume that the attacker has access to the RAG retriever and provide examples to support this assumption. However, this is a strong assumption, as in practice it is difficult, if not impossible, for an attacker to access the parameters or query the retriever. The authors should not assume that the attacker has this knowledge outright. Similar to paper [A], the authors could consider two scenarios: a black-box setting and a white-box setting. The attacks discussed in this paper align more with a white-box setting. Therefore, the authors should also include experiments that reflect a black-box setting.
2) The paper lacks essential baselines. It seems feasible to adapt the attacks from [A] to the context described in this paper. The authors should include important baseline comparisons.
3) During the retrieval phase, only k relevant passages are retrieved, but the default value of k used in the paper is not specified.
4) There are various potential defenses against the proposed attacks, such as paraphrasing. The authors should also evaluate the attack’s performance when such defense mechanisms are employed.


[A] PoisonedRAG Knowledge Poisoning Attacks to Retrieval-Augmented Generation of Large Language Models.

---

> ### Author Response · Authors · 2024-11-23
> **Reply to Reviewer ufqz (1/2)**
>
> We appreciate the reviewer's positive feedback on the clarity of our paper and the comprehensive experiments demonstrating the effectiveness of our proposed attack.
>
> > **Q1. Accessing to the RAG retriever is difficult.**
>
> **A:** We appreciate the reviewer's feedback and would like to address the concerns regarding the threat model. Access to retriever parameters is not only feasible but increasingly common, given the performance and availability of open-source retrievers. For instance, 28 open-sourced retrievers currently outperform proprietary models like OpenAI’s `text-embedding-3-large` across 213 test datasets on the Massive Text Embedding Benchmark (MTEB) Retrieval Leaderboard[2]. This indicates a huge market for deploying open-sourced retrievers, underscoring the potential impact of attacks targeting them. BadRAG is designed to exploit this vulnerability by creating poisoned passages tailored to these widely used retrievers. Moreover, the ease and low cost of distributing content on platforms such as Wikipedia and Reddit further facilitate the dissemination of these poisoned passages. This capability significantly enhances the reach and impact of attacks, posing a very real and substantial risk to realistic RAG systems.
>
> > **Q2. Comparison with PoisonedRAG.**
>
> **A: [Qualitative Comparison]:** We acknowledge that the black-box setting of PoisonedRAG, where the attacker does not need access to the retriever's internals, offers practical advantages. However, PoisonedRAG’s methodology—crafting poisoned passages for predefined specific **queries**—limits its attack impact. In contrast, BadRAG crafts poisoned passages for **triggers**, enhancing both flexibility and impact. It allows poisoned passages to be retrieved by any query containing predefined keywords (triggers), expanding the scope of potential attacks. Our technique, Merged COP, improves this flexibility further by enabling a single poisoned passage to respond to a range of related triggers, such as "Democratic Party" and "Biden". This means that queries like "Analyze Biden's policy" and "How do you evaluate the policies introduced by the Democratic Party?" would both successfully retrieve the same poisoned passage, enhancing the potential impact of the attack across different query formulations.
>
> **[Quantitative Comparison]:** We conducted experiments across three datasets using the Contriever under a black-box setting, with 10 poisoned passages injected. We evaluated the attack success rate for inputs both with and without triggers. The results are presented in the table below:
>
> |   Methods   | Queries |       |   NQ   |        |       |  MS  |        |       | SQuAD |        |
> |:-----------:|:-------:|:-----:|:------:|:------:|:-----:|:----:|:------:|:-----:|:-----:|:------:|
> |             |         | Top-1 | Top-10 | Top-50 | Top-1 | Top-10| Top-50 | Top-1 | Top-10| Top-50 |
> | PoisonedRAG |  Clean  |   0   |  0.02  |  0.51  |   0   |  0.03 |  0.18  |   0   |  0.01 |  0.07  |
> |             | Trigger |  0.08 |  2.18  |  6.24  |  0.12 |  3.04 |  7.69  |  0.05 |  1.62 |  5.26  |
> |    BadRAG   |  Clean  |  0.21 |  0.43  |  1.92  |  0.05 |  0.12 |  1.34  |  0.19 |  0.54 |  1.97  |
> |             | Trigger |  98.2 |  99.9  |   100  |  98.7 |  99.1 |   100  |  99.8 |   100 |   100  |
>
> BadRAG significantly outperforms PoisonedRAG with higher ASR for triggered inputs. This superiority is expected, given that PoisonedRAG targets specific queries, whereas BadRAG utilizes a backdoor paradigm that activates through any relevant trigger, providing a broader and more effective attack mechanism.
>
> While we recognize PoisonedRAG as a pioneering study on RAG system vulnerabilities, BadRAG offers a fundamental view of attack with a backdoor paradigm. Although BadRAG assumes access to a white-box retriever—a stronger assumption but realistic—it results in a greater and more realistic impact. We believe it can inspire the community to consider deeper, systemic vulnerabilities within RAG systems.

---

> ### Author Response · Authors · 2024-11-23
> **Reply to Reviewer ufqz (2/2)**
>
> >**Q3. The default value of K**
>
> **A:** Thanks for your reminder. The default value of k is set to 10, we have updated the description in our paper. The generator of RAG, unless otherwise specified, accepts the top-10 relevant retrieved passages as contexts.
>
> >**Q4. Discussion about potential defenses such as paraphrasing**
>
> **A:** In our original manuscript Appendix H, we evaluated the robustness of BadRAG against existing potential defenses proposed in Zhang et al. [1] and proposed a new potential defense against our BadRAG in Section 5. Additionally, we adopt the reviewer's suggestions to evaluate using paraphrasing as a defense method, and the analysis is posted in Appendix H. In short, the paraphrasing has little defense effectiveness, because the poisoned passage in BadRAG is optimized for a set of trigger words, which are semantically similar. Consequently, even if paraphrasing alters the trigger word, the substituted words are likely to still fall within the set of triggers, ensuring the retrieval of the poisoned passage.
>
> [1] Zhong et al. Poisoning Retrieval Corpora by Injecting Adversarial Passages
>
> [2] https://huggingface.co/spaces/mteb/leaderboard

---

### Official Review · Reviewer_ewGr · 2024-11-05

**Soundness:** 2
**Presentation:** 3
**Contribution:** 1
**Rating:** 3
**Confidence:** 5

**Summary:**

This paper proposes BadRAG as an attack on RAG. It is identified that poisoning several customized content passages could achieve a retrieval backdoor, where the retrieval works well for clean queries but always returns customized adversarial passages for triggered queries. Triggers and adversarial passages can be highly customized to implement various attacks. Utilizing contrastive optimization, BadRAG generates adversarial passages activated only by specific triggers. DOS and sentiment steering attacks are also explored.

**Strengths:**

1. This paper focuses on an important area, the security and safety of RAG.

**Weaknesses:**

1. This paper performs adversarial attack using backdoor settings. Concepts in this paper is misused.
2. The method used is adversarial attack, but it need to be introduced in the training phase, rather than the traditional test phase direct attack.
3. The former related work is nor discussed. [TrojanRAG: Retrieval-Augmented Generation Can Be Backdoor Driver in Large Language Models.]

**Questions:**

See weaknesses.

---

> ### Author Response · Authors · 2024-11-23
> **Reply to Reviewer ewGr**
>
> We appreciate the reviewer's recognition of the importance of the topic studied in our paper.
>
> > **Q1. The use of concept "backdoor".**
>
> **A:** Thanks for your insightful question. We respectfully disagree with the misused concept of "backdoor". We believe the BadRAG belongs to the backdoor attack because it follows the principle "inject a backdoor that can be later activated by a pre-defined trigger." BadRAG creates and injects the poisoned passages into the RAG corpus as a backdoor during the deployment phase, and uses the triggered quires to activate it later in the test phase, influencing the LLMs output. Unlike the traditional backdoor attacks that install the backdoor into the model, we reveal that other components besides the model can also be the driver of backdoor trojan for a complex ML system like RAG. Moreover, BadRAG can customize the trigger, while adversarial perturbations are obtained by optimization instead of pre-defining. Based on the above points, we believe the concept of "backdoor" is not misused in our paper.
>
> > **Q2. The method used is adversarial attack, but it needs to be introduced in the training phase, rather than the traditional test phase direct attack.**
>
> **A:** We agree that adversarial attacks are indeed applied during the test phase to get adversarial perturbation. However, as previously discussed, BadRAG is characterized as a backdoor attack. BadRAG employs an optimization-based method to craft poisoned passages specifically designed to be retrieved only by queries containing a predefined trigger, not by others. These poisoned passages are injected during the deployment phase and activated during the test phase, confirming that BadRAG adheres to the essence of backdoor attacks, which is to "inject a backdoor that can be later activated by the pre-defined trigger."
>
> > **Q3. Comparison with TrojanRAG.**
>
> **A:** We appreciate the reviewer's kind reminder to compare with another concurrent attack against the RAG system. In our original manuscript, we briefly discussed TrojanRAG, and we are glad to delve into a more detailed comparison and update our manuscript accordingly.
>
> Although both are backdoor attacks against the RAG, they have fundamentally different threat models and methods. TrojanRAG involves not only injecting poisoned passages into the user’s corpus but also requires the victim to deploy the trojaned retriever from the attacker. In contrast, BadRAG removes the need for the victim to use a poisoned retriever from the attacker. It creates poisoned passages that target any open-source retriever, making any user of these retrievers a potential victim. So BadRAG is very practical in real-world scenarios, especially given the availability of numerous good open-source embedding models that can be integrated with tools like Langchain and LlamaIndex to construct an RAG pipeline.

---

### Note · Authors · 2024-12-16

I have read and agree with the venue's withdrawal policy on behalf of myself and my co-authors.